# Sodium, Potassium and Iodine Intake, in a National Adult Population Sample of the Republic of Moldova

**DOI:** 10.3390/nu11122896

**Published:** 2019-11-28

**Authors:** Lanfranco D’Elia, Galina Obreja, Angela Ciobanu, Joao Breda, Jo Jewell, Francesco P. Cappuccio

**Affiliations:** 1World Health Organization Collaborating Centre for Nutrition, University of Warwick, Coventry CV4 7AL, UK; lanfranco.delia@unina.it; 2Department of Clinical Medicine and Surgery, “Federico II” University of Naples, 80131 Naples, Italy; 3Department of Social Medicine and Health Management, State University of Medicine and Pharmacy Nicolae Testemitanu, 2004 Chişinău, Moldova; galina.obreja@gmail.com; 4World Health Organization Country Office, 2012 Chişinău, Moldova; ciobanua@who.int; 5World Health Organization European Office for Prevention and Control of Noncommunicable Diseases, 2100 Copenhagen, Denmark; jewellj@who.int; 6World Health Organization European Office for Prevention and Control of Noncommunicable Diseases, 229994 Moscow, Russia; rodriguesdasilvabred@who.int; 7United Nations Children’s Fund, UNICEF, New York, NY 10017, USA; 8Division of Health Sciences, Warwick Medical School, University of Warwick, Coventry CV4 7AL, UK

**Keywords:** Republic of Moldova, salt, sodium, potassium, iodine, population

## Abstract

In the Republic of Moldova, more than half of all deaths due to noncommunicable diseases (NCDs) are caused by cardiovascular disease (CVD). Excess salt (sodium) and inadequate potassium intakes are associated with high CVD. Moreover, salt iodisation is the preferred policy to prevent iodine deficiency and associated disorders. However, there is no survey that has directly measured sodium, potassium and iodine consumption in adults in the Republic of Moldova. A national random sample of adults attended a screening including demographic, anthropometric and physical measurements. Sodium, potassium and iodine intakes were assessed by 24 h urinary sodium (UNa), potassium (UK) and iodine (UI) excretions. Knowledge, attidues and behaviours were collected by questionnaire. Eight-hundred and fifty-eight participants (326 men and 532 women, 18–69 years) were included in the analysis (response rate 66%). Mean age was 48.5 years (SD 13.8). Mean UNa was 172.7 (79.3) mmoL/day, equivalent to 10.8 g of salt/day and potassium excretion 72.7 (31.5) mmoL/day, equivalent to 3.26 g/day. Only 11.3% met the World Health Organization (WHO) recommended salt targets of 5 g/day and 39% met potassium targets (>90 mmoL/day). Whilst 81.7% declared limiting their consumption of processed food and over 70% not adding salt at the table, only 8.8% looked at sodium content of food, 31% still added salt when cooking and less than 1% took other measures to control salt consumption. Measures of awareness were significantly more common in urban compared to rural areas. Mean urinary iodine was 225 (SD: 152; median 196) mcg/24 h, with no difference between sexes. According to WHO criteria, 41.0% had adequate iodine intake. Iodine content of salt table was 21.0 (SD: 18.6) mg/kg, lower in rural than urban areas (16.7, SD = 18.6 vs. 28.1, SD = 16.5 mg/kg, *p* < 0.001). In most cases participants were not using iodised salt as their main source of salt, more so in rural areas. In the Republic of Moldova, salt consumption is unequivocally high, potassium consumption is lower than recommended, both in men and in women, whilst iodine intake is still inadequate in one in three people, although severe iodine deficiency is rare. Salt consumed is often not iodised.

## 1. Introduction

Non-communicable diseases (NCDs) are the leading causes of death globally [1] and their reduction is a health priority [2], with reduction in population salt consumption a cost-effective policy option (‘best buys’) [3]. In the Republic of Moldova, NCDs are the leading causes of death, and cardiovascular disease (CVD) represents the main cause of population morbidity and mortality, accounting for every second death in 2016 [4]. High blood pressure (BP) and unhealthy diets are major causes CVD in the world and account for most of the disease burden in the Republic of Moldova [5]. High salt in the diet (i.e., sodium chloride, 1 g = 17.1 mmoL of sodium) causes high BP, a high risk of vascular diseases [6,7,8,9,10] and other adverse health effects [11,12,13]. A lower salt intake reduces BP [7,8], cardiovascular events and overall mortality [9,10].

The World Health Organization (WHO) currently recommends for adults a consumption not higher than 5 g of salt daily [14]. However, in most countries in the world this recommendation is unmet [15,16,17]. Salt enters our diet not only as added salt to food and cooking by the consumer, but, in the Western diet, more often from processed food, food prepared in restaurants and other food outlets [18,19]. There is no direct estimate of population dietary salt intake in Republic of Moldova. However, it is likely to be high, as in neighbouring countries like Serbia (9.85 g/day) [20], Slovenia (11.3 g/day) [21] and Montenegro (11.6 g/day) [22]. In the Republic of Moldova it is a common habit to add salt to food at the table and when cooking, as well as eating processed food that have high salt content. In 2013 a national survey indicated that 24.3% of those surveyed always or often added salt to food, and 32.4% always or often ate processed foods that are high in salt [5]. Salt reduction strategies in the European region, including the Republic of Moldova, include monitoring and evaluation actions as one of their pillars [23].

In contrast to sodium, dietary potassium has beneficial effects on BP and cardiovascular health [24,25,26]. The Republic of Moldova lacks data on actual potassium consumption. The WHO currently recommends that adults should consume not less than 90 mmoL of potassium daily [27]. Finally, in the Republic of Moldova the prevention of iodine deficiency disorders recommends universal salt iodization [28]. Starting in 2009, the Ministry of Health authorised the production and placing on the market of iodized bottled water additionally to iodized salt. Since more than 90% of iodine consumed is excreted in the urine within 24–48 hours [29,30], 24 h urinary iodine excretion is a good marker of recent iodine intake and an ideal biomarker for estimating iodine status [31] in the entire adult population.

The aim of the present study was to establish current baseline average consumption of sodium, potassium and iodine by 24h urine collection, in a national random sample of men and women. The study also aimed to explore knowledge, attitudes and behaviour towards dietary salt.

## 2. Materials and Methods

### 2.1. Participants and Recruitment

A total of 1307 randomly selected men and women participated in the survey. They were all aged 18–69 years. They comprised residents of all Districts and Administrative Territorial Units ‘Gagauz-Yeri’, along with Chişinău and Bălti Municipalities. The survey did not cover the Districts from the left bank of the Nistru River and the Municipality of Bender (Figure 1). A probabilistic master sample from the National Bureau of Statistics’ Household Budget Survey was used to select the sample for the survey which was extracted in three phases: 150 Primary Sampling Units (PSU—communes, cities or sectors within cities) were selected; list of households from PSU were drawn; eligible individuals from households were identified. Random sampling proportional to size were stratified by sex, geography (north, centre, south and Chişinău), urban/rural, size of cities.

From the sampling frame and according to PAHO/WHO and EMRO-WHO Protocols [32,33], we excluded the following groups: pregnant women, individuals with heart failure, severe kidney disease, stroke, liver disease, people who had started diuretic therapy in the last two weeks, any other conditions that would compromise the collection of 24 h urine samples or a reliable informed consent.

The survey took place between 21st July and 5th September 2016. From the 1307 participants interviewed, 858 (66%) provided data for inclusion in the analysis. Thirteen had missing data, 263 admitted missing more than one void, 77 provided either under-collections (<23 h) or over-collections (>25 h) and 37 had urinary creatinine excretion outside two standard deviations (SDs) of the sex-specific distribution of urinary creatinine in the sample (Figure 2).

The survey was carried out in accordance with the Declaration of Helsinki and Good Clinical Practice [34]. Ethical approval for the survey was obtained from the Committee of Research Ethics of the National Agency for Public Health of the Republic of Moldova and participants provided written informed consent to take part.

### 2.2. Data Collection

The full methodology is reported in the Appendix A. In brief, there were three stages: (a) questionnaire, (b) physical measurements and (c) 24 h urine collections.

The questionnaire (adapted version of the WHO STEPS Instrument for NCD Risk Factor Surveillance) [35] obtained demographic and socio-economic status, frequency of salty food, fruit and vegetable consumptions, knowledge attitudes and behaviour on dietary salt.

Anthropometry, blood pressure (BP) and heart rate were measured in all participants with standardized protocols and validated equipment, as also described elsewhere [22,32,33]. Hypertension is defined as systolic and/or diastolic BP ≥ 140/90 mmHg or regular antihypertensive treatment [36].

After detailed and careful instructions (Text S1), participants provided a 24 h urine collection [32,33]. Sodium, potassium and creatinine determinations were carried out immediately [37,38]. Sodium and potassium concentration in the urine samples were determined using a Ion Selective Electrode with a Beckman Coulter Synchron CX5PRO system (High Wycombe, UK) and expressed in mmoL/L [37]. Creatinine concentration was determined through the Creatinine (urinary) Jaffé kinetic method and expressed in mg/dL [38]. These determinations were carried out at the ICS Medical Laboratory Synevo SRL in Chişinău. Urinary iodine was measured separately at the National Agency for Public Health of the Republic of Moldova using the ammonium persulfate digestion method with spectrophotometric detection by Sandell–Kolthoff reaction, expressed as mcg/L [39]. Iodine determinations in table salt were carried out by the titration method [40].

### 2.3. Statistical Analysis

With a standard deviation of 75 mmoL/24 h (alpha = 0.05, power = 0.80) in urinary sodium excretion, the study was designed to detect a reduction in salt consumption over time of around 1 g per day (~20 mmoL sodium/24 h). Considering an attrition rate of 50%, we aimed to select 240 participants per age and sex group [32,33]. The population was stratified in groups by sex (men and women), age (18–29 years, 30–44 years, 45–59 years, 60–69 years) and urban/rural areas. Therefore, 1920 individuals were originally needed to be selected (total *n* = 120 × 8 groups/0.5 attrition = 1920). T-test for unpaired samples or analysis of variance (ANOVA) was used to test differences between groups. The chi-square test was used for categorical variables. To convert urinary output into dietary intake, the urinary excretion of sodium (UNa) or potassium (UK) in mmoL/day were first converted to mg/day (for sodium 1 mmol = 23 mg of sodium, for potassium 1 mmol = 39 mg). The conversion from dietary sodium (Na) intake to salt (NaCl) intake was made by multiplying the sodium value by 2.542. Then, sodium values were multiplied by 1.05 (assuming that aproximately 95% of sodium ingested is excreted) [41]. For potassium dietary intake was calculated assuming 80% of the potassium ingested is excreted in the urine [42]. Urinary iodine was expressed in mcg/day. We used the cut-off targets for iodine consumption set by WHO (based on urinary iodine concentrations in mcg/L derived from 24h collections) [31]. Statistical analyses were carried out using SPSS, version 20 (SPSS Inc., Chicago, IL, USA). The results were reported as mean (SD and/or 95% CI) or as percentages, as appropriate. Two-sided *p* below 0.05 were considered statistically significant.

## 3. Results

The final population sample included 858 participants between 18 and 69 years old (*n* = 326 or 38% men and *n* = 532 or 62% women), recruited nationally (Figure 1).

### 3.1. Characteristics of the Participants

Mean age was similar in men and women, but men were taller and heavier than women and had a higher systolic BP (Table 1). The point prevalence of hypertension was 45.5% (385/858), comparable in men (148/326 or 45.8%) and women (237/532 or 45.2%; *p* > 0.05).

### 3.2. Daily Urinary Excretions of Volume, Sodium, Potassium and Creatinine and Salt and Potassium Intake

Urinary volume excretion was, on average, 1441 mL per day, higher in men than women, and higher in urban than rural areas (Table 2). Urinary creatinine excretion was 11.7 mmol per day, higher in men than women, but lower in urban than rural areas (Table 2). Mean urinary sodium was 172.7 (SD 79.3, median 161.9) mmoL/24h (Table 2), equivalent to a mean consumption of 10.8 (4.9) g of salt per day (Table 2). Men excreted more sodium than women (mean difference 18.1 mmoL/24h, *p* < 0.001), equivalent to ~1.1 g of higher salt consumption than women. WHO recommended levels of 5 g or less were met by just 97 participants (11.3%), with no difference between sex and area of residence. Mean urinary potassium was 72.7 (SD 31.5, median 68.8) (Table 2), equivalent to a mean consumption of 3.40 (1.47) g of potassium per day (Table 2).

Men excreted more potassium than women. Thirty-nine per cent of particiants met the levels of potassium excretion of 90 mmoL/day or more recommended by the WHO, with no difference between sexes and areas of residence.

### 3.3. Daily Intake of Iodine and Use of Iodised Salt

Urinay iodine excretion (as measure of intake) was adequate in 40.9% of participants, irrespective of sex or area of residence (Table 3). Iodine consumption was above requirement or excessive in 30.3% of the participants, irrespective of sex or area of residence. Of the 28.6% who fell into the category indicating insufficient consumption (equally distributed by sex or area of residence), only 2.3% had severe deficiency (Table 3).

Average urinary iodine excretion was 225 (SD: 152, median 196) mcg per day (Table 4), with no difference between sexes or areas of residence. Iodine salt content was, on average 21.0 (18.6) mg/kg, with no difference between men and women. However, participants in rural areas consumed table salt with significantly less iodine concentrations than those samples consumed in urban areas (*p* < 0.001; Table 4).

There were weak correlations between the amount of sodium excreted in the urine and the amount of excreted iodine in men and women or rural and urban settings (Figure 3).

There were also weak correlations between the amount of urinary iodine excreted in a day and the amount of iodine present in the table salt sampled from the households of individual participants (Figure 4).

### 3.4. Knowledge, Attitude and Behaviours Towards Salt Intake

Knowledge, attitude and behaviours toward the consumption of salt was assessed by asking participants about the frequency, quantity and type of salt used in the household, as well as their cooking habits and their attitudes towards dietary salt. A total of 35.4% of respondents mentioned that they added salt always or often before or while eating. The percentage of men who added salt always or often to their meal was significantly higher than that of women (47.8% vs. 27.7%; *p* < 0.001). A total of 61.3% of respondents reported that they always or often added salt when cooking or preparing food at home; this was the case more often in rural than in urban areas (69.8% vs. 47.5%; *p* < 0.001). More than half of the respondents (64.4%) mentioned that they used iodised salt when cooking or preparing food at home. Consumption of iodised salt, however, was higher in urban than in rural areas (86.1% vs. 52.9%; *p* < 0.001). About a quarter (26.7%) felt they consumed too much or far too much salt, women being more likely (32.1% vs. 23.3%; *p* < 0.01). More than half (67.2%) acknowledged that consuming too much salt could cause serious health problems; however, only 28.2% considered lowering salt intake in the diet to be very important. More than a quarter of respondents (27.8%) mentioned that they consumed processed foods high in salt, with more men than women doing so (34.9% vs. 23.5%; *p* < 0.001), and more often in urban than rural settings (39.2% vs. 20.8%; *p* < 0.001).

Participants were asked about actions they take to control salt intake on a regular basis. A total of 81.7% limited their consumption of processed food high in salt (Table 5). A total of 22.3% would use spices rather than salt, one in three would not add salt when cooking. Only 8% looked at salt/sodium content on food labels and 14.3% bought alternatives to salt. One in three avoided eating food prepared outside home and 0.8% took any other measure to reduce salt intake.

## 4. Discussion

This is the first national survey on sodium, potassium and iodine consumption ever carried out in adults in the Republic of Moldova, using the gold standard measure of 24 h urinary sodium, potassium and iodine excretions as biomarkers of intake. The results show unequivocally that salt consumption is high, potassium consumption is lower than recommended, both in men and in women. Furthermore, iodine intake is still inadequate in one in three people, although severe iodine deficiency is rare. However, universal salt iodization cover is still inadequate in many households both in urban and rural areas, where the use of iodized salt is still limited in the Moldovan diet.

### 4.1. Salt Consumption

Average salt intake was nearly 11 g per day, over two-fold of the WHO recommended maximum population target of 5 g per day [14]. Only 11.3% of the participants met the WHO salt targets. Men excreted more sodium than women, and in rural areas salt consumption was higher than in urban areas. Discretionary use of salt is common in the Republic of Moldova, with a third of participants adding salt regularly to food and half also using it regularly when cooking. The majority of participants knew that high salt causes serious health problems. However, only less than half thought it would be useful to reduce its consumption, and even fewer felt their own intake was not excessive and were doing anything to reduce it. The answers to these questions reveal an insufficient level of knowledge of the problem associated with high salt consumption amongst the participants and the unreadiness to transfer this knowledge to behavioural changes in using discretionary salt.

### 4.2. Potassium Consumption

Average population potassium intake was estimated at around 3.26 g per day, still lower than the WHO recommended minimum population target of 3.51 g per day (equivalent to 90 mmol per day [27]. Between 31% and 50% met WHO potassium targets. Men ate more potassium than women, likely due to the larger body size and volume of food eaten, rather than the quality of it. Salt and potassium are expressed as total quantities rather than consumption per calorie intake, hence the gender difference is mainly explained by the larger body size of men compared to women and the corresponding total food consumption compared to women. No difference in potassium intake was detected between rural and urban areas.

### 4.3. Iodine Consumption

Average daily iodine consumption was 225 mcg per day, with no difference between sexes or areas of residence. Severe iodine deficiency (<20 mcg/L according to WHO criteria [31]) was rare. However, more than a quarter had levels below 100 mcg/L (insufficient), and a quarter had levels either above requirement or excessive (above 300 mcg/L). The Republic of Moldova has adopted for a long time a policy of universal salt iodization for the control of iodine deficiency disorders [28]. It should be mentioned that as from 2009 the production and placing on the market of iodized bottled water was authorised by the government, which may have contributed to the increasing iodine supply, especially among more affluent population groups. By measuring the iodine content of the table salt used in the households visited for the screening, we were able to detect a significant lower iodine content in rural compared to urban areas (16.7 vs. 28.1 mg/kg). Moreover, the percentage of households with no iodized salt was greater in rural than urban areas (30.9% vs. 9.8%; *p* < 0.001). These results seem to suggest a variety of barriers, including possibly deterioration of iodized salt before reaching the users, reduced access, lower use, lack of awareness, costs, lack of use of iodized salt in food preparation by local producers, street vendors and the food industry. Another finding in our study was the lack of strong relationships between urinary sodium and iodine excretions and between urinary iodine excretion and iodine content in households’ table salt. These findings may, in part, indicate that some salt in diet derives from non-iodized sources. Assuming that the food eaten in the households is prepared with iodized salt (since the country has a national policy of universal salt iodization), a major component of the salt consumed may derive from food eaten outside the household prepared with non-iodized salt. It is also possible that the use of iodized bottled water has become an important source of iodine, not captured in the correaltions with iodized salt.

### 4.4. Comparison with Other European Countries

Our main findings show that the salt intake in the Republic of Moldova is as high or higher than that reported in many other European and neighbouring countries, both in men and women. In the recent MINISAL study in Italy, the daily salt intake of Italians was 10.9 g for men and 8.5 g for women [43], with large variations by region and socio-economic status [44]. In Northern Greece average intakes were 11.1 and 9.1 g per day for men and women, respectively [45]. In the national survey of salt consumption in Slovenia men ate 13.0 g and women 9.9 g per day [21]. Recently, in the city of Podgorica, in Montenegro, salt consumption was measured at 13.9 g in men and 9.9 g per day in women [22]. In the SALTURK II survey in Turkey, men consumed 15.7 g of salt per day and women 14.0 g per day, with higher salt consumption in rural compared to urban areas [46]. Finally, in Portugal a national survey as estimated the consumption of salt at 10.7 g per day in men and 10.2 g per day in women [47]. Potassium intake in the Republic of Moldova was lower than in Portugal [47] and higher than that measured in Italy [43], Greece [45], and Montenegro [22]. Men eat more potassium than women.

### 4.5. Strengths and Limitations

Our study included a large random sample of men and women representative of the Republic of Moldova. Salt and potassium intake were measured using the gold-standard method of 24 h urine collections [33]. We applied a rigorous quality control and used a highly standardized protocol to ensure completeness of urine collections, and a strict protocol of inclusion only of those fulfilling the quality control criteria, such as length of collection time and urinary creatinine excretion, markers of the accuracy of the collection. Our study is one of few studies having carried out at the same time a population based evaluation of daily iodine excretion in an adult population using 24 h urinary iodine excretion as a biomarker (rathe than spot urine samples), to assess the iodine status of a group of individuals who, whilst being supplemented with the universal salt iodization program, is not usually included in the population monitoring and surveillance on the effects of such policy.

Selection bias remains a possibility that we cannot rule out. We excluded a third of the participants as a result of the robust quality control for completeness of urine collections. Participants not delivering complete urines had lower weight, BMI, waist and hip circumferences and lower diastolic BP than those complying. No other differences were seen in their general characteristics (Appendix A). Urinary sodium and potassium excretions were only assessed once. Whilst we cannot characterise an individual’s intake [48], there is less likelihood that group estimates be biased. Finally, the absence of measurements of thyroid hormones does not allow a full assessment of the impact of both insufficient and excessive iodine intake on the adult population.

### 4.6. Impact and Policy Implications

The population of the Republic of Moldova is of just over 4 million, of whom approximately 75% over the age of 25 years. According to national health statistics from 2016, the mortality rate from diseases of the cardiovascular system is 617.3 per 100,000 population [49]. To meet the 30% reduction in population salt consumption set by WHO, the Republic of Moldova should aim at a 3.24 g per day reduction nationally. This reduction would be expected to avert 7.9% CVD events and 10.7% strokes every year, approximately 1460 CVD deaths per year.

The Republic of Moldova adopted the National Strategy on Prevention and Control of Noncommunicable Diseases 2012–2020 and its Action Plan [50]. Part of this pledge is to continue on the awareness campaigns already in place and to establish a comprehensive strategy involving legal measures, mandatory reformulation, nutritional labelling, efficient enforcement and good leadership [28]. Furthermore, a feasibility study of implementation and evaluation of essential interventions for the prevention of CVD in primary healthcare is currently under way in the Republic of Moldova, with a view towards a national scale-up [51].

The evidence of the level of sodium and potassium intake in the Republic of Moldova provides robust evidence to support action and to facilitate evaluation. Awareness, attitudes and behaviours about salt and its implication for health suggest that there is an intensification of public awareness campaigns and health promotion to improve the take up of preventive strategies aiming at reducing salt consumption, whilst at the same time increasing potassium intake by encouraging higher consumption of potassium-rich food. Awareness about hidden salt in processed food should be highlighted. The national program for reducing salt intake in the Republic of Moldova needs a multisectoral collaborative approach including not only public awareness and behaviour-change communication (including via health care professionals), but, more importantly, structured programs for reformulation that set the framework for the food industry to reduce salt in bread and bakery products and processed foods, major source of salt intake.

### 4.7. Conclusions

The present study provides valuable insights into ways to improve and adapt the universal salt iodization program. From one hand our results suggest that there are improvements to be made for a comprehensive take up of the policy nationally. On the other hand it confirms that both iodization and salt reduction policies are fully compatible, as agreed in a WHO Consensus Statement [31] and more recently confirmed in case studies in Italy [52] and China [53] where a moderate salt reduction in unlikely to compromise iodine status. Our data provides a useful baseline against which to monitor the impact of future initiatives.

## Figures and Tables

**Figure 1 nutrients-11-02896-f001:**
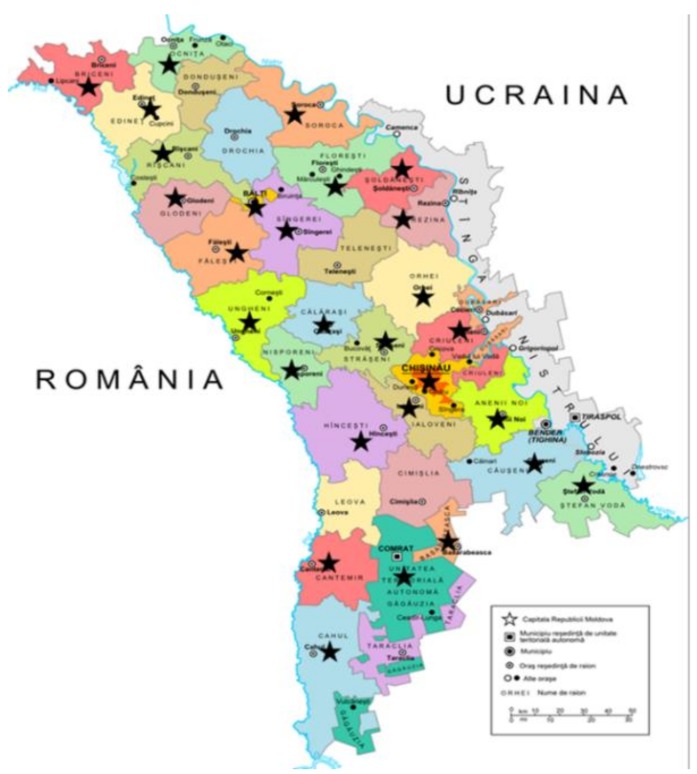
Geographical sampling from the Republic of Moldova. National proportional random sampling from 28 (marked with a star) of 37 Districts and Administrative Territorial Units ‘Gagauz-Yeri’, along with Chişinău and Bălti Municipalities. The sampling was as follows: Anenii Noi (1.3%), Balti (0.8%), Basarabeasca (1.4%), Briceni (4.7%), Cahul (3.5%), Călăraşi (2.4%), Cantemir (2.4%), Căuşeni (0.8%), Chişinău (30.7%), Comrat/ATU ‘Gagauz-Yeri’ (4.4%), Criuleni (4.3%), Edineț (3.1%), Făleşti (2.4%), Floreşti (2.2%), Glodeni (1.2%), Hînceşti (0.7%), Ialoveni (4.4%), Nisporeni (3.0%), Ocnița (2.7%), Orhei (4.8%), Rezina (1.7%), Rîşcani (0.6%), Sîngerei (1.9%), Șoldaneşti (2.6%), Soroca (2.2%), Ștefan Vodă (0.6%), Străşeni (2.7%) and Ungheni (6.3%).

**Figure 2 nutrients-11-02896-f002:**
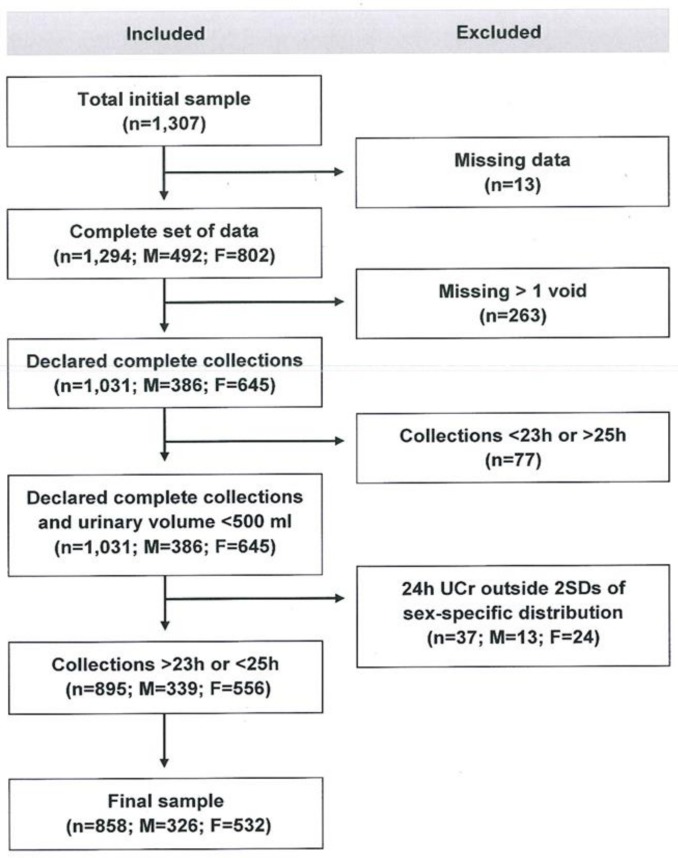
Stepwise procedure for the selection of valid participants according to protocol adherence, quality control and completeness of 24 hour urine collections.

**Figure 3 nutrients-11-02896-f003:**
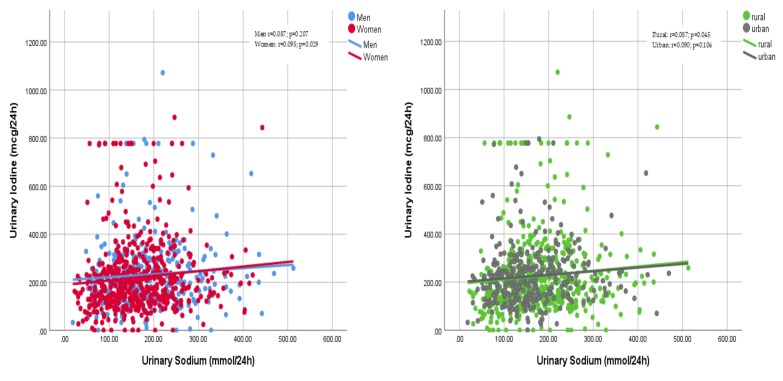
Correlations between urinary sodium and urinary iodine excretions by sex (left) and areas of residence (right). Left: men r = 0.087, *p* = 0.207, women r = 0.095, *p* = 0.029; right: rural r = 0.087, *p* = 0.045, urban r = 0.090, *p* = 0.106.

**Figure 4 nutrients-11-02896-f004:**
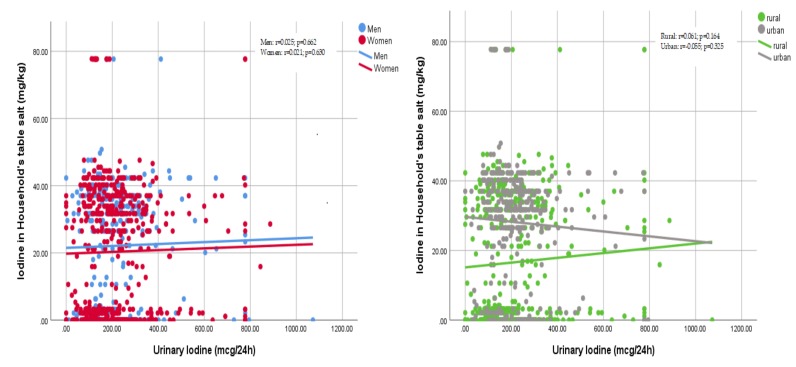
Correlations between urinary iodine excretions and iodine content of household’s table salt by sex (left) and areas of residence (right). Left: men r = 0.025, *p* = 0.662, women r = 0.021, *p* = 0.630; right: rural r = 0.061, *p* = 0.164, urban r = −0.055, *p* = 0.325.

**Table 1 nutrients-11-02896-t001:** Characteristics of the participants.

Variable	All	Men	Women
	(*n* = 858)	(*n* = 326)	(*n* = 532)
Age (years)	48.5 (13.8)	47.3 (13.6)	49.2 (13.9)
Height (cm)	166.7 (8.8)	172.8 (8.1)	162.9 (7.0) ^†^
Weight (kg)	78.2 (15.8)	82.0 (15.8)	75.8 (15.3) ^†^
B.M.I. (kg/m^2^)	28.1 (5.4)	27.4 (4.9)	28.6 (5.7) ^‡^
Waist circumference (cm)	–	93.8 (15.5)	91.8 (15.1)
Hip circumference (cm)	–	100.5 (12.3)	106.5 (14.0)
Systolic BP (mmHg)	134.3 (21.2)	136.1 (18.5)	133.1 (22.6) *
Diastolic BP (mmHg)	86.8 (11.9)	87.1 (10.8)	86.6 (12.6)
Pulse rate (b/min)	76.2 (9.5)	78.0 (10.3)	75.2 (8.8)
Hypertension ^#^ *n* (%)	385 (45.5)	148 (45.8)	237 (45.2)

Results are mean (SD) or as percentage; ^†^
*p* < 0.001; ^‡^
*p* = 0.002; * *p* = 0.04 vs. men. ^#^ SBP ≥ 140 mmHg and/or DBP ≥ 90 mmHg or on anti-hypertensive medications.

**Table 2 nutrients-11-02896-t002:** Daily urinary excretions of volume, sodium, potassium and creatinine and estimates of salt and potassium intake.

Variables	All	Men	Women	Rural	Urban
	(*n* = 858)	(*n* = 326)	(*n* = 532)	(*n* = 531)	(*n* = 327)
Volume (mL/24 h)	1441 (529)	1505 (536)	1401 (521) ^^^	1333 (427)	1616 (624) ^#^
Sodium (mmoL/24 h)	172.7 (79.3)	183.9 (86.0)	165.8 (74.1) ^†^	180.4 (80.2)	160.1 (76.2) ^#^
Salt intake (g/day)	10.8 (4.9)	11.5 (5.4)	10.3 (4.6) ^#^	11.3 (5.0)	10.0 (4.8) ^#^
Potassium (mmoL/24 h)	72.7 (31.5)	76.0 (33.4)	70.7 (30.1) *	73.8 (31.6)	71.0 (31.2)
Potassium intake (g/day)	3.40 (1.47)	3.55 (1.56)	3.31 (1.41) *	3.45 (1.47)	3.32 (1.46)
Creatinine (mmol/24h)	11.7 (5.0)	13.3 (5.6)	10.7 (4.2) ^#^	12.3 (4.8)	11.4 (5.0) ^†^

Results are mean (SD). ^#^
*p* < 0.001; ^^^
*p* < 0.005; ^†^
*p* < 0.01; * *p* < 0.02 vs. men or vs. rural

**Table 3 nutrients-11-02896-t003:** Proportions of participants meeting WHO targets for iodine consumption (based on urinary iodine concentrations in mcg/L derived from 24 h collections).

Group (mcg/L)	All	Men	Women	Rural	Urban
	(*n* = 858)	(*n* = 326)	(*n* = 532)	(*n* = 531)	(*n* = 327)
	*n* (%)	*n* (%)	*n* (%)	*n* (%)	*n* (%)
Insufficient (<100)	245 (28.6)	95 (29.1)	150 (28.2)	104 (31.8)	141 (26.6)
Severe (<20)	20 (2.3)	6 (1.8)	14 (2.6)	4 (1.2)	16 (3.0)
Moderate (20–49)	60 (7.0)	24 (7.4)	36 (6.8)	28 (8.6)	32 (6.0)
Mild (50–99)	165 (19.2)	65 (19.9)	100 (18.8)	72 (22.0)	93 (17.5)
Adequate (100–199)	351 (40.9)	132 (40.5)	219 (41.2)	131 (40.1)	220 (41.4)
Above requirement (200–299)	152 (17.7)	59 (18.1)	93 (17.5)	58 (17.7)	94 (17.7)
Excessive (≥300)	108 (12.6)	40 (12.3)	68 (12.8)	34 (10.4)	74 (13.9)

Results are number (%).

**Table 4 nutrients-11-02896-t004:** Daily urinary excretions of iodine and iodine content of household salt samples.

Variables	All	Men	Women	Rural	Urban
	(*n* = 858)	(*n* = 326)	(*n* = 532)	(*n* = 531)	(*n* = 327)
Iodine (mcg/24 h)	225 (152)	232 (154)	221 (150)	225 (145)	224 (128)
Iodine in table salt (mg/kg)	21.0 (18.6)	22.1 (18.2)	20.3 (18.9)	16.7 (18.6)	28.1 (16.5) ^#^

Results are mean (SD). ^#^
*p* < 0.001 vs. rural.

**Table 5 nutrients-11-02896-t005:** Knowledge, attitudes and behaviour towards the consumption of salt.

Participants Who:	All	Men	Women	Rural	Urban
	(*n* = 858)	(*n* = 326)	(*n* = 532)	(*n* = 531)	(*n* = 327)
Limit their consumption of processed food	81.7	79.7	82.4	80.4	83.9
Look at salt/sodium content in foods	8.8	10.1	8.2	3.8	17.2 *
Buy low salt/sodium alternatives	14.3	17.4	13.2	3.8	24.7 *
Do not add salt at the table	77.3	69.6	80.2	75.9	79.6
Do not add salt when cooking	31.1	24.6	33.5	13.3	61.3 *
Use spices instead of salt when cooking	22.3	15.9	24.7	25.3	17.2
Avoid eating food prepared outside a home	33.1	27.5	35.2	43.7	15.1 *
Take other measures to control salt intake	0.8	1.4	0.5	1.3	0

Results are expressed as % of column total. * *p* ≤ 0.001 vs. rural by Fisher’s exact test.

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
