# Peer review of "Sodium, Potassium and Iodine Intake, in a National Adult Population Sample of the Republic of Moldova"

_nutrients, 2019, doi:10.3390/nu11122896_

Round 1

Reviewer 1 Report

This is an interesting paper which outlines the results of a survey of sodium, potassium and iodine intakes and behaviours in adults in Moldovia.  The article is well well written, and suitable for Nutrients journal.  I have made some specific comments below:

The Abstract is too long, and much longer than the recommended maximum word count of 200 words. 

Lines 72-73- I would say that most (rather than several) countries do not meet the WHO recommendation for salt intake.  Please also clarify that in a Western diet most salt consumed in processed food.  This is not the case for all populations.

Line 163- please provide a reference for the assumption that 95% of ingested sodium is excreted in the urine. 

Line 164- Please provide a reference or references for the assumptions that 70%, 85% or 95% of potassium ingested is excreted in the urine.

Line 174- The phrase “men and women were of the same age” suggests that they were matched by age.  Do you mean this or do you mean that mean age for men and women was similar?

Line 184 and 189- the phrase “normally distributed with a tail toward higher values” is somewhat contradictory.  It would be useful to report the mean and median to justify this.

Table 2- reports two estimates of potassium intake, yet the legend stats that three different assumptions are made.  Further I presume the figures in the legend are incorrect and should be “assuming Ur.K is 70%, 86% or 95% of intake”.  Which of these assumptions are reported in the table?

Discussion- it would be useful to have some discussion of how the different assumptions of urinary potassium excretion have been interpreted, and the implications of this for the findings of the study.

Author Response

Reviewer #1

The Abstract has been cut substantially, although it is still more than 200 words (page 1) Added ‘most’ and clarified the Western diet (page 2) Ref 41 provided We have now reviewed the evidence and decided to settle for a single estimate provided by the INTERMAP Study in Western populations where the difference between urinary potassium and dietary potassium was approx. 80% (Ref 42, A119 to A122). We therefore, recalculated the estimated potassium intake from the potassium excretion using a factor of 1.2. Text and tables have been edited accordingly The phrase has been modified as suggested (page 5) The phrase has been removed and median values have been added to the text (page 6) Table 2 now reports a single value for potassium intake (see point 4 above) We have simplified the potassium issue referring to INTERMAP estimate (REF 42). No longer do we apply different assumptions

Reviewer 2 Report

The prevention of cardiovascular diseases remains a priority to date, by taking into consideration the fact that the mortality rates for these diseases remain high in many parts of the world. The manuscript is of great interest and well organized. Some aspects could further improve the quality of the article and that deserve to be evaluated. Here are some of my suggestions:

- The summary is very comprehensive but verbose. I recommend resizing it to give a more concise overview of the work

- Considering the close correlation between sodium and potassium, it could mention the hypothetical utility of evaluating the ratio of assumption between these two elements. Potassium intake can mitigate the effects of sodium excess

- Why was urinary excretion not normalized to creatinine excretion? It would have been a method to discriminate between individual changes in kidney function.

- In paragraph 3.1,  the greater weight and height of men is highlighted, specifying also the differences in systolic pressure compared to women. Why the contrast of higher systolic and lower BMI between men and women is not mentioned?

- At line 189 I believe you refer to urinary potassium and not sodium

- In discussing the validity of urinary iodine excretion to define its intake, I believe we should pay attention to the fact that it is a system of estimation and not a direct measurement. Its validity is, in fact, of moderate and not high relevance.

- The difference between iodine in table salt available in urban areas compared to rural areas is not sufficiently discussed. Does the Moldovan republic lack uniformity in fortification methods? What is this difference due to?

- Images 3 and 4 should be provided at a higher resolution. For example, it seems to me that the correlation between urinary sodium and iodine loses significance in men but in women it is very high and relevant. I cannot read the significance of the correlation between the population of urban areas.

- At line 219 a weak correlation is alleged but in the legend of figure 4 the value of p indicates that there is no significance of the correlation

- Is there a possibility that it can be analyzed the statistical difference between urinary iodine /sodium in respondents who report consuming salt or foods with added salt or using iodized salt compared to others, respectively?

- Is there an interaction between sodium and potassium excretion?

- Table 5 should be more adequately be positioned before the discussion

- Among the limitations, there should be mentioned the absence of dosage of thyroid hormones and therefore the possible physiological influence of insufficient intake of iodine cannot be detected

- At line 325 there is too much punctuation (two subsequent dots)

- From the last paragraph, the authors could extrapolate conclusions to be separated from the discussion

- In the supplementary material, I believe that "Miotype" is actually "myotape"

- Paragraph 2.3 of the supplementary material could be removed to complete the session in materials and methods in the manuscript because it is redundant.

Author Response

Reviewer #2

The Summary has been shortened significantly The interaction between sodium and potassium in contributing to blood pressure reductions and CVD prevention is beyond doubt. However, it was not the scope of the present study to assess its combined ‘effect’ on outcomes. The present study hjad no statistical power to look at those aspects (see also answer to 10). Sodium intake is expressed as total intake, not related to calorie intake or body mass, as other nutrients. Differential excretion of sodium relative to creatinine, whilst altering the capacity of some equation to predict 24-hour urinary sodium excretion from a single-void collection, do not provide a valid solution for the assessment of salt consumption within population groups (Nowson CA et al. J Clin Hypertens 2019; on-line July 8; doi: 10.1111/jch.13725) The study was powered for the detection of group mean sodium excretions. We have refrained from undertaking statistical analyses on ‘individuals’ as our sample size would be too small to detect differences in other variables. Furthermore, a proportion of participants were on anti-hypertensive medications as this would prevent us from analysing blood pressure values Corrected – thank you Re-phrased on page 2 We have rephrased the discussion on iodine and table salt on page 9-10, acknowledging possible explanations for our findings The reviewer is correct in seeing some statistical significant correlation (urinary sodium and iodine in women and in rural settings). However the correlation coefficients are still <0.1, when we would expect almost complete concordance if all the iodine has been provided through iodized salt. Hence, a weak association Figure legend now reports all correlations in detail As explained above (point 4) we have refrained from undertaking analyses on individual relationships as the study is no powered for that and single urine collections are not adequate to characterize individuals. In doing so we follow recent recommendations from several international scientific organizations (Campbell NRC, et al. J Clin Hypertens 2014; 16(7): 469-71; Campbell NRC, et al. J Clin Hypertens 2019; 21: 700-709; Cappuccio FP, et al. Nutr Metab Cardiovasc Dis 2019; 29: 107-114; Cappuccio FP and PS Sever. J Hum Hypertens 2019; 33: 345-348) Table 5 has been relocated at the best possible We have added a sentence to address the issue, as suggested (page 10). Thank you – corrected A new Conclusion paragraph has been created as suggested Typo corrected in the Supplementary material Paragraph 2.3 has been reduced removing repetitive information

Round 2

Reviewer 1 Report

The manuscript has been revised and is now suitable for publication.

Reviewer 2 Report

The authors responded to the comments. The manuscript was modified according to the required revisions.

This manuscript is a resubmission of an earlier submission. The following is a list of the peer review reports and author responses from that submission.